# Deconstructing the Regularization of Batch-Norm

**Yann N. Dauphin**
Google Research
ynd@google.com

**Ekin D. Cubuk**
Google Research
cubuk@google.com

## Abstract

Batch normalization (BatchNorm) has become a standard technique in deep learning. Its popularity is in no small part due to its often positive effect on generalization. Despite this success, the regularization effect of the technique is still poorly understood. This study aims to decompose BatchNorm into separate mechanisms that are much simpler. We identify three effects of BatchNorm and assess their impact directly with ablations and interventions. Our experiments show that preventing explosive growth at the final layer at initialization and during training can recover a large part of BatchNorm's generalization boost. This regularization mechanism can lift accuracy by $2.9\%$ for Resnet-50 on Imagenet without BatchNorm. We show it is linked to other methods like Dropout and recent initializations like Fixup. Surprisingly, this simple mechanism matches the improvement of $0.9\%$ of the more complex Dropout regularization for the state-of-the-art Efficientnet-B8 model on Imagenet. This demonstrates the underrated effectiveness of simple regularizations and sheds light on directions to further improve generalization for deep nets.

## 1 Introduction

Deep learning has made remarkable progress on a variety of domains in the last decade. While part of this progress relied on training larger models on larger datasets, it also depended crucially on the development of new training methods. A prominent example of such a development is batch normalization (BatchNorm) (Ioffe and Szegedy, 2015), which has become a standard component of training protocols. For example, state-of-the-art models in image recognition (Szegedy et al., 2017; He et al., 2016; Tan and Le, 2019), object detection (He et al., 2019; Du et al., 2019), and image segmentation (Chen et al., 2017) all use BatchNorm. Despite its prominence, the mechanisms behind BatchNorm's effectiveness are not well-understood (Santurkar et al., 2018; Bjorck et al., 2018; Yang et al., 2019).

Perhaps at the core of the confusion is that BatchNorm has many effects. It has been correlated to reducing covariate shift (Ioffe and Szegedy, 2015), enabling higher learning rates (Bjorck et al., 2018), improving initialization (Zhang et al., 2019), and improving conditioning (Desjardins et al., 2015), to name a few. These entangled effects make it difficult to properly study the technique. In this work, we deconstruct some of the effects of BatchNorm in search of much simpler components. The advantage of this approach compared to previous work is that it allows going beyond correlating these effects to BatchNorm by evaluating their impact separately. The mechanisms we consider in this work are purposefully simple. These simpler mechanisms are easier to understand and, surprisingly, they are competitive even at the level of the state-of-the-art.

Our contributions can be summarized as follows:

1. **How does normalization help generalization?** We isolate and quantify the benefits of the different effects of BatchNorm using additive penalties and ablations. To our knowledge, we are the first to provide empirical evidence that BatchNorm's effect of regularizing against explosive growth at initialization and during training can recover a large part of its generalization boost. Replicating this effect with Fixup initialization and the proposed additive penalty improves accuracy by $2.9\%$ for Resnet-50 without BatchNorm.

2. **Links to Fixup and Dropout.** We draw novel connections between the regularization on the final layer, Dropout regularization, Fixup initialization and BatchNorm.

3. **Simplicity in regularization.** The mechanism we identify can be useful as a standalone regularization. It produces $0.9\%$ improvement on the Efficientnet B8 architecture, matching the more complex Dropout regularization.

## 2 DECOMPOSING THE REGULARIZATION EFFECTS OF BATCH NORMALIZATION

In this section, we break BatchNorm into different mechanisms that can be studied separately. BatchNorm (Ioffe and Szegedy, 2015) is a technique that accelerates training by standardizing the intermediate activations of a deep network. It achieves this standardization by using explicit normalization instead of relying on an additive regularization. While BatchNorm has been correlated to many effects, it is still unclear which effect, if any, explains most of its generalization boost.

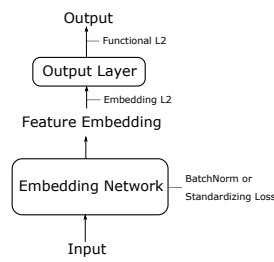

The effects we evaluate are the implicit regularizing effect on the norms at the final layer, and also its primary effect of standardizing the intermediate layers. In order to test these purposefully simple mechanisms we will rely on ablations and additive penalties. The use of additive penalties allows us to disentangle these effects where we control for the positive effect of BatchNorm on initialization by using the recently proposed Fixup initializer (Zhang et al., 2019).

Figure 1: Diagram of a neural network showing where the mechanisms operate.

### 2.1 REGULARIZING AGAINST EXPLOSIVE GROWTH IN THE FINAL LAYER

First, we characterize the implicit effect of normalization on the final layer. Consider a neural network of the form $\text{NN}(\mathbf{x}) = \mathbf{W}\text{Emb}(\mathbf{x})$ with loss $\mathcal{L}$ where $\mathbf{x} \in \mathbb{R}^I$ is the input of the network, $\mathbf{W} \in \mathbb{R}^{K \times H}$ is the final weight matrix in the model, and $\text{Emb}(\mathbf{x}) : \mathbb{R}^I \to \mathbb{R}^H$ is a feature embedding network with $L$ layers. Let us take the common case where $\text{Emb}(\mathbf{x}) = \text{Swish}(\gamma\text{BatchNorm}(\text{PreEmb}(\mathbf{x})) + \beta)$ where $\text{PreEmb}(\mathbf{x})$ is the output of a residual network, $\text{Swish}(\mathbf{x}) = \mathbf{x}\sigma(\rho\mathbf{x})$ is the Swish activation (Ramachandran et al., 2017; Elfwing et al., 2018) with scalar parameter $\rho$ (typically denoted $\beta$), and BatchNorm parameters $\gamma, \beta$. BatchNorm makes weight decay regularization on $\gamma, \beta$ approximately equivalent to an additive penalty on the norm on the feature embedding

$$\mathcal{L}(\text{NN}(\mathbf{x})) + \lambda\|\gamma\|^2 + \lambda\|\beta\|^2 = \mathcal{L}(\text{NN}(\mathbf{x})) + \frac{\lambda}{4}E[\|\text{Emb}(\mathbf{x})\|^2] + O(|\rho|). \tag{1}$$

See the Appendix A for the derivation. It means that the norm of the BatchNorm parameters alone is enough to directly control the norm of the feature embedding. It is a guarantee that the norm of the feature embedding cannot grow explosively during training as long as these parameters are small. This regularization effect of BatchNorm can occur even without explicit weight decay due to the tendency of stochastic gradient descent to favor low norm parameters (Wilson et al., 2017).

This equivalency does not hold without BatchNorm because the activation of the embedding network become an important factor in the norm of the feature embedding ($\|\gamma\|^2 + \|\beta\|^2 \neq E\left[\|\gamma\text{PreEmb}(\mathbf{x}) + \beta\|^2\right]$ in general). Indeed, (Balduzzi et al., 2017; Gehring et al., 2017; Zhang et al., 2019) have shown that the activations of residual networks without BatchNorm tend explode exponentially in the depth of the network at initialization. This results in an extremely large embedding norm, even though the parameters are relatively small. We confirm experimentally in Section 4.3 that networks without BatchNorm have much larger feature embedding norms.

**Feature Embedding L2** (EL2) We propose to assess the effect of this regularization mechanism by isolating it as the following additive penalty

$$R_{EL2}(\text{NN}) = \frac{1}{H}E[\|\text{Emb}(\mathbf{x})\|^2]. \tag{2}$$

Adding this regularization to the loss allows us to test the impact of this mechanism independently on networks with and without BatchNorm. We dub this regularization embedding L2 for short in the following sections as it is L2 with a metric that is at the embedding network function level, with no additional penalties on the intermediate layers. It is applied right before the classification layer, in other words right after the final average pooling layer for residual networks. We will see with our experiments that this simple regularization can in large part recover the regularization boost of BatchNorm, has links to several known methods and is practically useful even at the level of the state-of-the-art.

**Functional L2** (FL2) Regularizing the feature embedding has an implicit effect on the final output norm ($E[\|\text{NN}(\mathbf{x})\|] \leq \text{\small 1/2}\|\mathbf{W}\|^2 + \text{\small H/2} \cdot R_{EL2}(\text{NN})$). In order to test the impact of this effect, we will also evaluate a direct penalty on the norm of the final output

$$R_{FL2}(\text{NN}) = \frac{1}{K}E[\|\text{NN}(\mathbf{x})\|^2].\tag{3}$$

We dub this regularization functional L2 for short in the following sections as it is L2 with a metric that is at the full network function level. It is applied on the logits of the model for classification. In Section 4.4, we investigate whether it is regularizing this norm or the feature embedding norm that is more closely correlated to better generalization.

### 2.2 STANDARDIZING THE INTERMEDIATE ACTIVATIONS OF THE MODEL

As a baseline mechanism, we will also consider an additive penalty that encourages the normalization of every intermediate layer called standardization loss (Collins et al., 2019). This is a useful reference to gauge the effectiveness of the regularization on the embedding described in the previous sub-section. It also helps disentangle the side-effects of normalization. The penalty is

$$D_{KL}(P(\mathbf{x})\|\mathcal{N}(\mathbf{x}|0, I)) = \frac{1}{2}\sum_i (\mu_i^2 + \sigma_i^2 - \log \sigma_i^2 - 1)\tag{4}$$

where $\mu_i$ is the mean and $\sigma_i$ is the variance of an intermediate layer of the network. These are the same statistic that are computed by BatchNorm and a penalty is added to the loss for all the intermediate layers. This regularization has been considered in (Collins et al., 2019). They found that it accelerated learning but fell significantly short of the generalization boost of batch normalization. However, they did not account for the positive effect of normalization on initialization. We correct for this in our experiments using the recently proposed Fixup initialization (Zhang et al., 2019).

## 3 DRAWING LINKS TO OTHER METHODS

In this section, we will draw connections between the mechanisms considered for BatchNorm and other methods.

### 3.1 DROPOUT REGULARIZATION

Dropout (Hinton et al., 2012) is a regularization that prevents overfitting by randomly omitting subsets of features during training. Despite its early popularity, its use has declined with the rise of batch normalized convolutional networks. Ghiasi et al. (2018), and Zagoruyko and Komodakis (2016), find that it produces comparatively much smaller improvements when applied to the intermediate layers of such networks. However, Tan and Le (2019) has shown that state-of-the-art results can be obtained by applying Dropout only at the input of the final layer of the network. Interestingly, we can relate this particular use of Dropout to BatchNorm. Wang and Manning (2013) have shown that Dropout with MSE loss can be isolated as an additive regularization when applied at the last layer

$$\mathcal{L}_{\text{Dropout}} = \frac{1}{N}\sum E\left[(\mathbf{W}\texttt{Dropout}(\texttt{Emb}(\mathbf{x}_i)) - y_i)^2\right]$$
$$= \frac{1}{N}\sum (\mathbf{W}\texttt{Emb}(\mathbf{x}_i) - y_i)^2 + \lambda\text{tr}(\mathbf{W}\text{diag}(E[\texttt{Emb}(\mathbf{x})\texttt{Emb}(\mathbf{x})^T])\mathbf{W}^T)$$

where the additive Dropout regularization is $R_{\text{Dropout}}(\text{NN}) = \text{tr}(\mathbf{W}\text{diag}(E[\texttt{Emb}(\mathbf{x})\texttt{Emb}(\mathbf{x})^T])\mathbf{W}^T)$ and $\lambda$ is the strength of the penalty. In this formulation, we can see that Dropout is related to the

mechanisms in Section 2.1 as follows

$$K \cdot R_{FL2}(\text{NN}) \approx R_{\text{Dropout}}(\text{NN}) \leq \frac{1}{4}\|\mathbf{W}\|^4 + \frac{H}{4}R_{EL2}(\text{NN}). \tag{5}$$

The approximate relationship to functional L2 relies on the assumption that the features are relatively decorrelated. To our knowledge this close but simple relationship between the expected norm of the output and Dropout with MSE loss had not been noted before. The upper bound with embedding L2 gives us a guarantee on Dropout robustness when training with BatchNorm and weight decay. In some sense, this means that networks with BatchNorm already incorporate a regularization effect similar to that conferred by Dropout. This can explain why networks with BatchNorm have tended to benefit relatively less from Dropout.

The approximate relationship to functional L2 can be extended to networks with cross-entropy by using Taylor expansion and assuming the network has low-confidence. This assumption is likely only correct at initialization. In comparison, a related upper-bound can be found for embedding L2 using Taylor expansion. We will see in Section 4.5 that embedding and functional L2 can match or exceed Dropout for state-of-the-art architectures on Imagenet such as Efficientnet (Tan and Le, 2019). This is surprising because using an additive penalty at the final layer is much simpler than Dropout.

## 3.2 FIXUP INITIALIZATION

Fixup (Zhang et al., 2019) is a recent initialization method for the ubiquitous residual networks. Recall that residual networks without normalization have explosive dynamics with conventional initializations (Balduzzi et al., 2017; Gehring et al., 2017; Zhang et al., 2019). In fact, the output scale of the networks grows exponentially with their depth $L$. This means that the embedding and functional L2 at initialization are

$$R_{EL2}(\text{NN}_{\text{NoBN}}^{t=0}), R_{FL2}(\text{NN}_{\text{NoBN}}^{t=0}) \in \mathcal{O}(2^L) \tag{6}$$

where $t$ is the training iteration. These explosive dynamics can be very detrimental to learning (Balduzzi et al., 2017). Fixup aims to stabilize the training of very deep residual networks without normalization. It does so by initializing the network so that the output does not change explosively in the first gradient step, that is $\|\text{NN}_{\text{Fixup}}^{t=1}(\mathbf{x}) - \text{NN}_{\text{Fixup}}^{t=0}(\mathbf{x})\| \in \Theta(\eta)$ where $\eta$ is the learning rate. They show this stabilizes learning and the output scale. We find that Fixup initialization minimizes the initial penalties when compared to conventional initialization

$$R_{EL2}(\text{NN}_{\text{Fixup}}^{t=0}), R_{FL2}(\text{NN}_{\text{Fixup}}^{t=0}) \in \mathcal{O}(1) \tag{7}$$

since it initializes residual branches to zero. Most importantly, Fixup ensures this does not grow explosively in the first few steps. Zhang et al. (2019) observe that Fixup can improve generalization compared to conventional initialization, but requires strong regularization to be competitive to networks with normalization. In Section 4.3, we will see that regularizing the functional L2 not just at initialization but throughout training remarkably improves generalization.

## 4 EXPERIMENTS

In this section, we evaluate the regularization mechanisms on a set of architectures for two challenging benchmarks.

## 4.1 DATASETS

**CIFAR-10** CIFAR-10 has 50k samples for training and 10k samples for test evaluation. We tune hyperparameters using 5k of the training samples as a validation set. We then train a final model using the whole training set of 50,000 samples with the best hyperparameters for 10 different seeds, and report the median test accuracy. Each model is trained on a single GPU. The robustness is reported as the accuracy on CIFAR-10-C dataset (Hendrycks and Dietterich, 2019), averaged over all corruptions and their severities.

**SVHN** SVHN has 73,257 samples for training and 26,032 samples for testing (note that we do not consider the 531,131 "extra" samples). We tune hyperparameters using 3,257 of the training samples

as a validation set. We then train a final model using the whole training set of 73,257 samples with the best hyperparameters and report the median test accuracy for 10 different random initializations.

**Imagenet** Imagenet is a large scale image recognition dataset with over 1.2M examples and 1000 classes. Following the past work, we report classification accuracy on the validation set of 50k examples.

## 4.2 IMPLEMENTATION DETAILS

**WideResnet** We train WideResNet-28-10 architecture (Zagoruyko and Komodakis, 2016) on CIFAR-10, which is one of the most commonly used architectures for this dataset. We train the model for 200 epochs using a cosine learning rate decay with a batch size of 128. We use the standard data augmentation of horizontal flips and pad-and-crop. For models without BatchNorm, we use Fixup initialization. We use the activation function Swish (Ramachandran et al., 2017), with the $\rho$ value initialized to 0.

**Resnet-50** Resnet-50 (He et al., 2016) has quickly become a standard architecture to evaluate on Imagenet. Our implementation makes use of the improvements proposed by Goyal et al. (2017). Only the networks trained without BatchNorm use Fixup initialization (Zhang et al., 2019). In order to improve results for the standardizing loss, we found it was useful to use a special case of Fixup where the residual branches are not initalized to zero: when the standardization loss is used we initialize the last layer of each residual block inversely proportional to the square root of the depth. The models are trained for 90 epochs with a batch size of $512$ and a precision of float32. All results are the average of two seeds. The coefficient for the standardizing loss is cross-validated in the range from $10^{-7}$ to $10^{-5}$ logarithmically. We found that higher coefficients led to divergence at high learning rates. The coefficient for embedding and functional L2 was found in the range from $0$ to $1$ with increments of $0.1$.

**Efficientnet** Efficientnet (Tan and Le, 2019) is the state-of-the-art architecture on Imagenet at the time of writing. We will evaluate on the biggest version trained with RandAugment (Cubuk et al., 2019b) data augmentation and without additional data, called the B8 (Xie et al., 2020). We follow the implementation of Tan and Le (2019) and the hyper-parameters they found optimal. We use early stopping based on a held-out validation set of 25022 images. All results reported are averaged over two random seeds.

## 4.3 RESULTS: ABLATION OF BATCHNORM AND INTERVENTIONS

Table 1: Accuracy on CIFAR-10 dataset for Wide-Resnet-28-10.

| Method | Accuracy (%) | Robustness (%) |
|---|---|---|
| Baseline (WRN-28-10 without BatchNorm) | $94.9 \pm 0.1$ | $82.8 \pm 0.1$ |
| Baseline + BatchNorm | $96.1 \pm 0.1$ | $82.8 \pm 0.2$ |
| Baseline + Standardizing Loss ($\lambda = 1e{-}7$) | $95.0 \pm 0.2$ | $83.4 \pm 0.2$ |
| Baseline + Functional L2 ($\lambda = 1.0$) | $95.5 \pm 0.2$ | $84.1 \pm 0.3$ |
| Baseline + Embedding L2 ($\lambda = 0.5$) | $95.5 \pm 0.2$ | $84.1 \pm 0.2$ |

Table 2: Accuracy on Imagenet dataset for Resnet-50. Functional L2 with proper initialization allows us to match the generalization of networks trained with batch normalization.

| Method | Accuracy (%) |
|---|---|
| Baseline (Resnet-50 without BatchNorm) | $73.1 \pm 0.1$ |
| Baseline + BatchNorm | $75.7 \pm 0.0$ |
| Baseline + Standardizing loss ($\lambda = 1e{-}5$) | $74.7 \pm 0.2$ |
| Baseline + Functional L2 ($\lambda = 0.4$) | $76.0 \pm 0.1$ |
| Baseline + Embedding L2 ($\lambda = 1$) | $76.0 \pm 0.0$ |

First, we measure the impact of the regularization mechanisms under consideration for networks where we have removed BatchNorm. We also include results for network trained with BatchNorm

for reference. On CIFAR-10 (Table 1) and Imagenet (Table 2), we observe a significant gap between the baseline network and the network trained with BatchNorm. The baseline networks benefit from improved initialization with Fixup, which (Zhang et al., 2019) find stabilizes training at higher learning rates and increases accuracy by up to 2.9% on Imagenet without BatchNorm. However, the baseline networks do not have extra regularization during training.

We find that the standardization regularization produces modest robustness improvements on CIFAR-10 but not for the accuracy, and moderate improvements on Imagenet. We see that this regularization does not replicate BatchNorm's generalization boost, even with proper initialization. This matches the results of Collins et al. (2019) which considered the use of this regularization without accounting for the initialization effects of BatchNorm. We do not believe this indicates that standardization is not an important effect but rather reflects a failure of this particular approach. There are a number of possible explanations, including that this type of constraint is difficult for the optimizer to balance successfully with the classification loss. Indeed, we see that only comparatively small regularization coefficients were found to produce stable training with high learning rates.

In comparison, we note that replicating standardization's implicit effect on the norms at the final layer produces better results on both CIFAR-10 and Imagenet. While it is enough to match the generalization boost of BatchNorm on Imagenet, this is not the case on CIFAR-10. However, the scale of the improvement suggest that controlling the norm of the output through the norm of the feature embedding is an important benefit of standardization and as a result BatchNorm. In particular, embedding and functional L2 produces an improvement of close to $3\%$ on Imagenet and $0.6\%$ on CIFAR-10. We analyze this results more closely in Section 4.4.

To further test the effect of output norm regularization on a small dataset, we experiment with the SVHN core set. The Wide-Resnet-28-10 model with BatchNorm achieves 96.9% test accuracy on this dataset, whereas the same model without BatchNorm (but with Fixup initialization) achieves 95.5%. If we utilize the Functional L2 or the Embedding L2 regularization, we reach 96.5% accuracy. Similar to the models on CIFAR-10, output norm regularization offers a significant improvement in generalization (1.0%) for a model without BatchNorm.

Bjorck et al. (2018) have shown that BatchNorm allows using larger learning rates without divergence. It is thus interesting to take note of how the optimal learning rate changed when we removed BatchNorm. We observe that in our experimental setup on Imagenet the optimal base learning rate with BatchNorm of 0.1 remains optimal with embedding and functional L2. In our experimental setup on CIFAR-10, the optimal learning rates increased from 0.1 with BatchNorm to 0.2 with functional L2.

While BatchNorm is effective at improving accuracy, we find that it does not improve the robustness of the baseline model to distortions applied at test-time. We measure the robustness of the models by evaluating their accuracy on the 15 different corruptions described by Hendrycks and Dietterich (2019), which include corruptions from noise, blue, weather, and digital categories. Models trained with standardizing loss have 1.1% lower accuracy than the models trained with BatchNorm, but their robustness is 0.6% higher. This suggests that while BatchNorm might be improving robustness due to the standardization of the intermediate layers, it is reducing robustness by a larger amount due to its other effects (negative effects of BatchNorm on adversarial robustness have been discussed by Galloway et al. (2019)). Functional L2 is able to improve both accuracy and robustness (by 1.3% over the baseline model and the BatchNorm model), by its ability to reduce the output norm.

### 4.4 Analysis: How much does this explain the regularization of BatchNorm?

In the previous section, we show that the mechanisms considered are effective. This section investigates their relationship to BatchNorm.

**How much does BatchNorm regularize the norm at the last layer?** Figure 2 (left column) shows learning curves for networks with and without BatchNorm. As the networks without BatchNorm use a suitable initialization, we observe that the networks converge at similar speeds at first. However, we see a generalization gap emerge later in training on both datasets. We see that this gap is associated with larger growth throughout training in the norm of the final output and feature embedding for networks without BatchNorm. This confirms that BatchNorm prevents explosive growth in the representation from cascading to the output during training. This also makes clear that these quantities

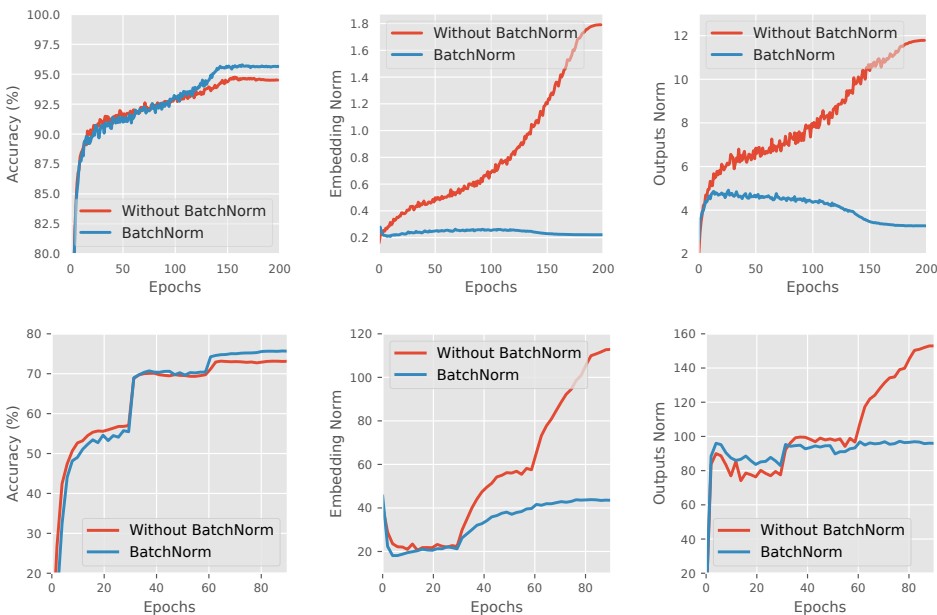

Figure 2: Statistics of models with and without BatchNorm on CIFAR-10 (top) and Imagenet (bottom) showing the accuracy (left), feature embedding norm (middle), mean output norm (right). Without additional regularization models without BatchNorm display explosive growth in the feature embedding norm. This correlates with poor generalization.

must be controlled throughout training and not just at initialization. And given that these networks are already trained with a well-tuned parameter L2, it is evidence that parameter L2 is not sufficient to control these quantities without BatchNorm.

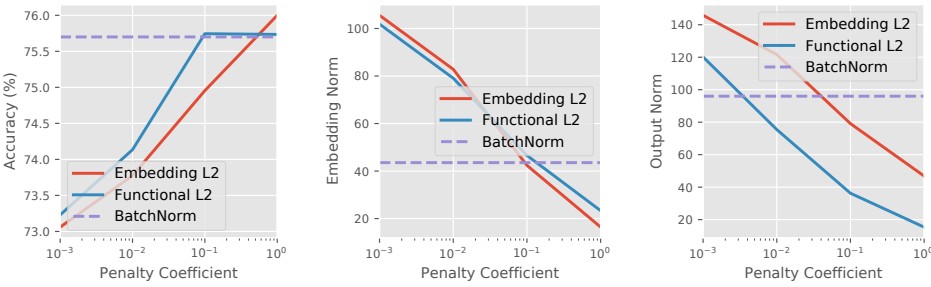

Figure 3: Effect of increasing the penalty coefficient for Imagenet on the accuracy (left), feature embedding norm (middle), and output norm (right). We see that embedding and function L2 recover most of BatchNorm's generalization boost when the embedding norm matches that of BatchNorm networks.

**Does regularizing the norm to the same level as BatchNorm improve results?** In Figure 3, we see that the accuracy increases with the coefficient of functional or embedding L2. Both the output and feature embedding norm reduce as expected. We see that at comparable output norms, networks with functional L2 perform much worse than BatchNorm. Indeed, the coefficient of functional L2 must be quite small to allow for such high norms. In contrast, we see that the methods perform comparably when the norm of the feature embedding is similar. These results indicate that the coefficient must be high enough to control the reduce the norm of the feature embedding as much as BatchNorm to obtain good results. This is consistent with the interpretation that the methods prevent explosions in the residual representation of the network from cascading to the output. Thus controlling the norm of the feature embedding appears more tightly related to generalization than the

norm of the final output. Like other regularizations such as classical L2, a single regularization does not tell the whole story and it matters how these quantities are minimized.

**Can we disentangle BatchNorm's effect on initialization to its effect on the norm at the last layer? Can we train networks with only a single BatchNorm layer at the feature embedding level?** Next, we evaluate Resnet-50 with only a single BatchNorm layer at the level of the final feature embedding on Imagenet (added before the average pooling layer). Let us first consider the case without Fixup initialization. This bad initialization leads to an explosion at the level before the BatchNorm layer, which has a norm of $1.8 \times 10^5 \pm 3.8 \times 10^4$ at initialization. Normally this network would not train at reasonable learning rates but thanks to the BatchNorm at that final layer it trains to $73.8 \pm 0.1$ accuracy. This a slightly higher accuracy than with Fixup only but lower than with BatchNorm in every layer. This shows that preventing explosion in the feature embedding is sufficient to allow training even without proper initialization. In combination with Fixup, the accuracy rises to $74.9 \pm 0.0$. This network contains over $100\times$ less normalization layers, but recovers nearly $70\%$ of the generalization improvement with respect to Fixup. It is corroboration of the previous results with additive penalties that proper initialization and controlling the feature embedding norm recovers a large part of the boost of BatchNorm.

Table 3: Accuracy drop from $50\%$ Dropout at test time for Resnet-50 models trained without Dropout on Imagenet. Networks regularized with BatchNorm or embedding L2 are considerably more robust to Dropout than the baseline.

| Method | Accuracy Drop (%) |
|---|---|
| Baseline | $-22.8 \pm 0.1$ |
| BatchNorm | $-11.0 \pm 0.3$ |
| Embedding L2 | $-18.5 \pm 0.1$ |

**How much do these methods increase natural robustness to Dropout?** Section 3.1 establishes a link between BatchNorm, embedding L2 and Dropout, with a link to functional L2 at initialization only. We showed that a low feature embedding norm should correlate to Dropout regularization. Table 3 shows the robustness to Dropout at test time for networks that were not trained with Dropout. We see that networks with embedding L2 suffer considerably less from Dropout corruption at test compared to the baseline. Similarly, the use of BatchNorm also boost robustness to Dropout. While it had been observed that Dropout regularizing during training is less effective for networks with BatchNorm, this is the first time it has been shown that BatchNorm networks have a naturally higher robustness to Dropout.

## 4.5 Regularization with BatchNorm

In this section, we evaluate using the regularizations on the final layer with BatchNorm. As explained in Section 2.1, BatchNorm already reduces the embedding L2 to some degree. However, BatchNorm and weight decay by themselves does not make it easy to control the strength of the regularization. On one hand, it would require using independent weight decay coefficients between parameters $\gamma$ and the other parameters, and on the other they only approximately reduce these quantities. Embedding L2 allows us to minimize this quantity directly.

Table 4 shows our results for Efficient B8. This model produces state-of-the-art results for Imagenet without additional data sources. We find that embedding and functional L2 boosts accuracy by $0.9\%$ compared to Efficientnet B8 without Dropout. This improvement matches that of Dropout. It is surprising that regularizing such a simple additive penalty improves results for a state-of-the-art architecture at the same level as the more complicated Dropout regularization. This is even more surprising considering that the baseline is already regularized by powerful methods like RandAugment (Cubuk et al., 2019b) and StochasticDepth (Huang et al., 2016). This suggests that further investigation of simple regularizations could be beneficial even for state-of-the-art models.

## 5 Conclusion

In this work, we have quantified a number of the regularization effects of BatchNorm. Surprisingly, our empirical study shows that preventing explosion of the feature embedding norm at initialization

Table 4: Accuracy on Imagenet dataset for Efficienet-B8.

| Method | Accuracy (%) |
|---|---|
| ResNet-50 (He et al., 2016) | 76.0 |
| DenseNet-169 (Huang et al., 2017) | 76.2 |
| ResNet-152 (He et al., 2016) | 77.8 |
| Inception-v3 (Szegedy et al., 2017) | 78.8 |
| ResNeXt-101 (Xie et al., 2017) | 80.9 |
| NASNet-A (Zoph et al., 2018) | 82.7 |
| AmoebaNet-C (Cubuk et al., 2019a) | 83.5 |
| GPipe (Huang et al., 2019) | 84.3 |
| Efficientnet B8 without Dropout | $84.3 \pm 0.0$ |
| Efficientnet B8 with Dropout at last layer | $85.1 \pm 0.0$ |
| Efficientnet B8 without Dropout + Functional L2 | $85.1 \pm 0.1$ |
| Efficientnet B8 without Dropout + Embedding L2 | $85.2 \pm 0.1$ |

and during training can recover a large part of the generalization improvement of BatchNorm. Furthermore, this much simpler regularization produces improvements on state-of-the-art architectures such as Efficienet-B8. We hope that these results will help guide the way to more effective regularization methods.

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

## A   APPENDIX

Here we provide the derivation for Equation 1. Consider the Taylor expansion of this expression around $\rho = 0$

$$\|\texttt{Swish}(\mathbf{x})\|^2 = \frac{1}{4}\|\mathbf{x}\|^2 + O(|\rho|). \tag{8}$$

Putting this together with the mean squared norm of the feature embedding

$$E[\|\texttt{Emb}(\mathbf{x})\|^2] = E[\|\texttt{Swish}(\gamma\texttt{BatchNorm}(\texttt{PreEmb}(\mathbf{x})) + \beta)\|^2], \tag{9}$$

we have

$$E[\|\texttt{Emb}(\mathbf{x})\|^2] = \frac{1}{4}E[\|\gamma\texttt{BatchNorm}(\texttt{PreEmb}(\mathbf{x})) + \beta)\|^2] + O(|\rho|). \tag{10}$$

This further simplifies due to the normalizing effect of BatchNorm

$$E[\|\gamma\texttt{BatchNorm}(\texttt{PreEmb}(\mathbf{x})) + \beta\|^2] = \sum_i E\left[(\gamma_i\texttt{BatchNorm}(\texttt{PreEmb}(\mathbf{x}))_i + \beta_i)^2\right] \tag{11}$$

$$= \|\gamma\|^2 + \|\beta\|^2, \tag{12}$$

yielding the result

$$E[\|\texttt{Emb}(\mathbf{x})\|^2] = \frac{1}{4}\|\gamma\|^2 + \frac{1}{4}\|\beta\|^2 + O(|\rho|). \tag{13}$$

