# OpenReview forum: "Deconstructing the Regularization of BatchNorm"
_ICLR.cc/2021/Conference — ICLR 2021 Poster_

### Official Review · AnonReviewer2 · 2020-10-27
**This paper aims to disentangle and study the regularization effect of batch normalization. By directly placing penalty on either the standardization of intermediate layers or the L2 norm of the output feature, the authors strive to match the performance boost from Batchnorm. Through experiments, the author came to the conclusion that it is the prevention of the explosive norm in the output feature that is the main contribution of batchnorm in network regularization.**

**Rating:** 4
**Confidence:** 4

**Review:**

** Overview
This paper is generally well written and well organized. It is easy to read. It strives to disentangle the two possible effects of batchnorm in neural network in order to study its main contribution to network regularization. The proclaimed two possible effects are
1. standardizing the intermediate activations
2. regularizing against explosive growth in the final layer.
By dropping the batchnorm layer and replace it with one of the two penalty functions --  penalty on normalization or penalty on the l2 norm of the final layer -- the authors claimed that much of the performance gain from batchnorm is recovered from the norm regularization of the final layer. Although the paper is interesting, I do not find the results to be very convincing (please see "questions" section for clarification)

** Pros
1 Well written and well organized.
2 Experiments are conducted to explain the disentangled effect of batchnorm in regularization.

** Cons and Questions.
1. Even though the batch norm layer has the effect of standardizing intermediate layers and limiting the size of the final layers, it is not necessarily true that their effects can be replicated by simply putting a penalty on the network loss during training. In particular, In table 1 and table 2, the authors use very small penalty coefficient lambda to replicate the "standardizing effect", and claimed that "we found that higher coefficients led to divergence at high learning rate." This leads me to think that the results from directly penalizing the effects of batchnorm are unconvincing.

2. One of the benefit of batchnorm is that it allows for a significantly larger learning rate. After dropping the batchnorm and putting the penalty on, does such benefit still exist? I would very much prefer to see that the maximal "allowable" learning rate does not decrease after dropping the batchnorm layer.

3. The authors claimed a connection between dropout and penalization of the output feature in equation (5). However, this plausible connection is only valid when features are decorrelated -- which seems to be a very strong assumption.

4. The authors also make a difference between "feature embedding L2" and "Functional L2", whose difference seems unnecessary when weight decay penalty is used because of equation (5). Is that correct?

5. If I understand correctly, it is the explosion of gradient instead of the output feature that makes training method hard to converge. Can the authors elaborate on the merit of limiting the feature size? After all, even if the feature is large, a rescaled weight matrix in the final layer can easily bring everything back to normal size.

---

> ### Author Response · Authors · 2020-11-12
> **Re: This paper aims to disentangle and study the regularization effect of batch normalization. By directly placing penalty on either the standardization of intermediate layers or the L2 norm of the output feature, the authors strive to match the performance boost from Batchnorm. Through experiments, the author came to the conclusion that it is the prevention of the explosive norm in the output feature that is the main contribution of batchnorm in network regularization.**
>
> Thank you for your comments. We are happy to hear that you thought the paper is well written and well organized. One general issue you raise is about the standardizing regularization, which is not our main contribution. We have only included this method as a baseline. Below we address the questions one by one.
> 1. Indeed, we agree that the standardizing regularization does not reach its desired effect. We included this method as a baseline, we report in the paper that this regularization term does not perform as well as BatchNorm and this was also reported by the original paper from Collins et al. In comparison, directly penalizing the embedding L2 provides more significant improvements and benefits from large coefficients. We will make this more clear.
> 2. This is a good point, and we already have results on this that we will add to the paper. Briefly, we do find that output-norm penalty increases the optimal learning rate as well as the allowable maximum learning rate. Our batch-norm baseline uses an optimal learning rate of 0.1 (in agreement with literature). Our baseline without batch-norm or output-norm penalty uses an optimal learning rate of 0.1 as well, and is unstable on larger learning rates. With the output-norm penalty however, the optimal learning rate is 0.2, and is mostly stable at 0.3 and even 0.4.
> 3. Our connection between dropout and the penalties considered is not only approximate, we have an upper bound that holds exactly on the right hand of Eq 5. Table 3 provides some experimental results supporting this. We hope this addresses this concern.
> 4. Indeed, as noted above equation 3 the feature embedding L2 bounds the functional L2  when combined with weight decay. We could have just experimented with feature embedding L2, but chose to also consider functional L2 to evaluate which effect was more closely related to generalization. Figure 3 shows that feature embedding L2 is most closely associated with better results.
> 5. While the scope of this paper is regularization, from an optimization perspective we can see the large output norms as decreasing the temperature on the softmax, which can lead to large gradients. And if the features grow explosively, in the scheme you describe the final weights would have to shrink proportionally close to 0 which can make learning difficult. In practice, we see in Figure 1 (right) that the networks do not naturally compensate for large features in this way and generalization suffers.
>
> We appreciate the helpful questions, and hope that we have addressed your concerns. Do you have any remaining concerns?

---

> ### Author Response · Authors · 2020-11-24
> **Rebuttal Revision**
>
> We have uploaded a revision of the paper that includes a discussion on the optimal learning rates - as you had requested. We look forward to your feedback on our comments below and these changes as well.

---

### Official Review · AnonReviewer1 · 2020-10-28
**Heuristic studies on the understanding the regularization of BN**

**Rating:** 6
**Confidence:** 3

**Review:**

The paper empirically studies the regularization of BN. It proposes the point that the BN's effect is connected with the regularizing against explosive growth in the final layer. To motivate this point, it takes a single-layer case and shows the BN approximately penalizes on the norm of the feature embedding thereon. Two regularizations are proposed according to the point and are used to justify it.

In the experiments, it combines the proposed regularizations with Fixup and tests on CIFAR-10 and ImageNet along with three typical architectures including, WideResNet, ResNet-50, and Efficientnet. The observed phenomena support the claims and further find that prevent the explosion of the norm can recover most of the improvements of BN. Also, the proposed regularizations achieve great performance on benchmarks via properly setting the coefficient.

The studied direction is very important and would bring some new understanding to the ICLR community. Motivated by the simple case and connections with the previous techniques, like dropout, fixup, standardizing are really good points. The reviewer can see clear differences and contributions to the literature. Experiments are conducted on large-scale benchmarks and different architecture. So the results and conclusions tend to be general. Overall the paper is easy to follow.

-----------------------------------------
Some concerns and comments are listed below.

The experiment would run a grid search to select a coefficient for the L2 regularization. Table 1,2 only list the performance of the best choice. The reviewer wonders how sensitive would the coefficient affect accuracy, especially on ImageNet. (The reviewer guesses the experiments of Figure 3 is not conducted on ImageNet.) If it is significant, the method may not suitable for common use.

In Sec. 4.4, the author claims that the generalization gap is associated with the growth in the norm. But, the gap shows in the very late phase, while the norm growth starts at the very beginning. There's no strong correlation.

In all experiments, the proposed regularizations perform based on the Fixup initialization. The reviewer wonders how the performance change if you change an initialization (e.g. Xavier Init), though the concept of your regularization is the same as the Fixup. To see if the proposed regularization can help training even without proper initialization, like the last layer BN experiments did in Sec.4.4. Also, the paper currently does not prove the importance of the initialization via any ablations but directly use the Fixup.

To sum up, the reviewer thinks the paper finds some points about the regularization effects of BatchNorm, but not much principal, and would rate 6.


-------------------after rebuttal----------
I thank the author for your answer. Here're the response to your latest reply.

For point 1, I am aware of your performance and encourage you to add the discussion in the main paper.

For point 2, I still doubt it since I think your claim may only hold for the ImageNet experiment on the Outputs Norm term. According to figure 2, the network does not faster converge with the help of BN. The regularization gap happened in the late stage. But, the gap of two networks on the feature embedding norm and mean output norm happens at the very beginning and keeps increasing. (except  ImageNet experiment on the Outputs Norm term)

Overall, I think the paper may bring some insights to understand the BN and would like to keep my original score.

---

> ### Author Response · Authors · 2020-11-12
> **Re: Heuristic studies on the understanding the regularization of BN**
>
> Thank you for a careful and overall positive review of our paper. We will use your comments and questions to improve the paper. Below we address the comments one by one.
> 1. We confirm that the setting of the coefficient is not very sensitive. This is confirmed by Figure 3 which was actually conducted on Imagenet. We will clarify that in the paper.
> 2. Regarding the correlation, it is best seen in Figure 3, which shows a strong correlation between the final norm and generalization (lower is  better) on Imagenet. Figure 1, which you reference, shows how the norm grows throughout training. It also shows a correlation between the final norms and the accuracy. We will clarify that we are referring to the norm at the end of training in 4.4.
> 3. We find this regularization helps even with other initializations. Also, the Fixup paper already included comparisons to Xavier initialization and found that it reached only 68.5%, which is lower than 73.1% accuracy we can reach with Fixup. We will mention that in the paper.
>
> We hope that your concerns are addressed, and we appreciate the suggestions. Do you have any remaining concerns?

---

> > ### Comment · AnonReviewer1 · 2020-11-24
> > **Response to authors**
> >
> > I thank you for your efforts and rebuttal. I still have some concerns about the mentioned points.
> >
> > **1**. According to Figure 3, the method has a 2% performance difference in ImageNet with different coefficients. I would not say it is non-sensitive, since if a trick can achieve 2% improvements on ImageNet which may potentially publish as a paper.
> >
> >
> > **2**. Here is the sentence I refer to in Sec 4.4. `However, we see a generalization gap emerge later in training on both datasets. We see that this gap is associated with significant growth in the norm of the final output and feature embedding for networks without BatchNorm. `  Refer to Figure 2, The growth in the norm of embedding starts at the very beginning, while the generalization gap happens in the very late stage. This may indicate a weak correlation between the final norms and accuracy, and oppose the proposed points.
> >
> > **3**. Hope you can add the results in the revision.

---

> > > ### Author Response · Authors · 2020-11-25
> > > **Response to further comments**
> > >
> > > **1** The optimal performance with functional L2 can be achieved for functional L2 parameters spanning an order of magnitude, as can be seen in Figure 3a (note the x axis is in log scale). We confirm that this is also true for CIFAR-10: optimal performance can be achieved for a wide range of functional L2 parameters (again about an order of magnitude). As you point out, for very small values of the functional L2 coefficient, the performance is not as good, which is to be expected. Similar plots can be seen for fundamental hyperparameters such as learning rate and weight decay. Our point was just that optimal performance does not require a cherry picked coefficient value, but values as small as 0.1 and as large as 1.0 give equally good performance. We will add the CIFAR-10 results in the paper as well.
> > >
> > > **2** It is common for overfitting to occur later in training and so it is similarly common for the benefit of regularization to be seen later in training. The benefit a regularization is hard to assess without finishing training and it is not common to judge regularizations solely by looking at points in the middle of training. Rather we see that increasing the regularization to reach smaller norms consistently reduces the gap in Figure 3. We will clarify this sentence in a revision.
> > >
> > > **3** Thank you for the questions and suggestions. We will certainly add the discussions and the results in the final revision.

---

> > > > ### Author Response · Authors · 2020-11-25
> > > > **Revision**
> > > >
> > > > We have uploaded a revision of the paper that clarifies the sentence you pointed out.

---

> ### Author Response · Authors · 2020-11-24
> **Rebuttal Revision**
>
> We have uploaded a revision of the paper that clarifies the caption of Figure 3 and adds more discussion on the use of Fixup. We look forward to your feedback on our comments below and these changes as well.

---

### Official Review · AnonReviewer3 · 2020-10-30
**Interesting paper with potentially significant observations**

**Rating:** 7
**Confidence:** 5

**Review:**

## Paper Summary

This paper aims to improve our understanding of BatchNorm's positive effects on training and generalization by identifying its constituent effects, and then replicating those effects with simpler techniques. Through a series of experiments using these simpler techniques, the authors show that perhaps the most important mechanism by which BatchNorm improves training, and especially generalization, is by preventing excessive growth of the activations of the layer before the output layer.

## Strengths

The paper makes a particularly strong case for studying the relationship between the "embedding norm" -- the norm of inputs to the final layer of the network -- and successful training/generalization. It does this by

a) showing that use of BatchNorm can implicitly control this norm, especially if weight decay is applied to BatchNorm parameters.

b) showing that in networks trained with BatchNorm, the value of this norm remains much lower than in those trained without BatchNorm.

c) showing that using simple L2 regularization penalty on this norm, its value can be prevented from growing and, in combination with Fixup initialization, networks without BatchNorm can achieve similar generalization performance as those with BatchNorm.

I think this constitutes a good amount of initial evidence that this technique may be useful for both researchers and practitioners.

## Weaknesses

I did not find major weaknesses, though it remains a little puzzling why all the clearly positive results with activation norm penalties matching or outperforming BatchNorm are only obtained on ImageNet. This indicates that perhaps more tasks/datasets should have been explored by the authors (the paper uses only two datasets).

## Review Summary

Despite the weakness above, I think the results are sufficiently interesting to be published and evaluated by the wider community. I recommend accepting this paper, though results on more diverse tasks and datasets will certainly improve this paper (and my rating).

## Minor comments

- Sec. 3.2: Authors say "Zhang et al. show residual networks without normalization have explosive dynamics with conventional initializations. In particular, they show that the output scale of the networks grows exponentially with their depth $L$". It is incorrect to attribute these to Zhang et al. instead of Balduzzi et al. Please correct this.
- Sec. 4.3: I don't see "modest improvements" with standardizing loss on CIFAR-10 in Table 1. Given the standard errors, I would say that there is effectively no improvement.
- Section 2.1 uses $\rho$ in Swish instead of $\beta$ which is mentioned in Sec. 4.2.
- Sec. 3.1: "Wang and Manning (2013) has" should be "Wang and Manning (2013) have"
- Table captions should be on top, not bottom of the tables, according to ICLR formatting guidelines.

---

> ### Author Response · Authors · 2020-11-19
> **Re: Interesting paper with potentially significant observations**
>
> Thank you for your thoughtful comments and for taking the time to provide detailed feedback. We will use this feedback to improve the paper. First, we will modify the paper to reflect the textual changes you suggested. And based on your feedback, we have started experiments on the SVHN dataset. Our results already show that the proposed regularizations can provide strong improvements on this dataset. In particular,  we found that a WRN-28-10, trained only on the core SVHN dataset (the set of ~73k samples) gets 95.5% with Fixup initialization. If we add functional L2 regularization, we find that the accuracy increases to 96.3%. We have only run a single experiment so far and have not been able to tune the parameters very well, so we expect further improvements can be achieved with a few more experiments. We will be adding these experiments to the paper as we finalize them.

---

> ### Author Response · Authors · 2020-11-24
> **Rebuttal Revision**
>
> We have uploaded a revision of the paper that addresses all the textual changes you suggested. We will add SVHN experiments to the final version of the paper as we finalize the experiments. We look forward to your feedback on our comments below and these changes as well.

---

### Author Response · Authors · 2020-11-24
**Rebuttal Revision**

We would like to thank the reviewers for their valuable comments. We have uploaded a revision of the paper that incorporates some of your suggestions. We look forward to your feedback on our comments below and these changes as well.

---

### Decision · Program_Chairs · 2021-01-07
**Final Decision**

**Decision:**

Accept (Poster)

**Comment:**

This paper analyzes BatchNorm through a series of experiments and shows that BatchNorm improves training and generalization by preventing excessive growth of the activations of the previous to the last layer. The paper received mixed reviews. Two reviewers find that the paper brings clear and important new contributions to the understanding of BatchNorm, and appreciate the experimental evaluation, though some suggestions for improving the experiments were also provided. The other reviewer appreciated the contribution of regularizing against explosive growth in the previous to the last layer, but did not find the results to be very convincing as they can limit the learning rate. The authors responded that, contrary to the reviewer's claim, the learning rate is not limited. Overall, while there are some aspects that need improvement, and the authors should address those in the final version, this is a solid paper that brings an interesting contribution to ICLR.